# Cardiorenal Syndrome: An Updated Classification Based on Clinical Hallmarks

**DOI:** 10.3390/jcm11102896

**Published:** 2022-05-20

**Authors:** Rainer U. Pliquett

**Affiliations:** 1Department of Nephrology and Diabetology, Carl-Thiem Hospital Cottbus, 03048 Cottbus, Germany; r.pliquett@ctk.de; 2Department of Internal Medicine, University Hospital Halle, Martin-Luther University Halle-Wittenbeg, 06108 Halle (Saale), Germany

**Keywords:** cardiorenal syndrome, chronic heart failure, chronic kidney disease, acute kidney injury, acute heart failure

## Abstract

Cardiorenal syndrome (CRS) is defined as progressive, combined cardiac and renal dysfunction. In this mini review, a historical note on CRS is presented, the pathomechanisms and clinical hallmarks of both chronic heart failure and chronic kidney disease are discussed, and an updated classification of CRS is proposed. The current consensus classification relies on the assumed etiology and the course of the disease, i.e., acute or chronic CRS. Five types are described: type-I CRS presenting as acute cardiac failure leading to acute renal failure; type-II CRS presenting as chronic cardiac failure leading to chronic renal failure; type-III CRS presenting as acute kidney injury aggravating heart failure; type-IV CRS presenting as chronic kidney failure aggravating heart failure; and type-V CRS presenting as concurrent, chronic cardiac and renal failure. For an updated classification, information on the presence or absence of valvular heart disease and on the presence of hyper- or hypovolemia is added. Thus, CRS is specified as “acute” (type-I, type-III or type-V CRS) or “chronic” (type-II, type-IV or type-V) CRS, as “valvular” or “nonvalvular” CRS, and as “hyper-” or “hypovolemia-associated” CRS if euvolemia is absent. To enable the use of this updated classification, validation studies are mandated.

## 1. Introduction

From a physiological perspective, the definition of cardiorenal syndrome (CRS) is viewed as combined cardiac and renal dysfunction aggravating the failure of both organs progressively. Currently, the consensus classification of cardiorenal syndrome (CRS) is based on the assumed etiology and the course of disease, i.e., an acute or chronic course of CRS. Five CRS types are described: type-I CRS presenting as acute cardiac failure leading to acute renal failure; type-II CRS presenting as chronic cardiac failure leading to chronic renal failure; type-III CRS presenting as acute kidney injury aggravating heart failure; type-IV CRS presenting as chronic kidney failure aggravating heart failure; and type-V CRS presenting as concurrent acute or chronic cardiac and renal failure or injury [1]. CRS type-V represents its own category, as non-cardiac and non-renal conditions such as diabetes mellitus or sepsis are the offending causes. The bidirectional relationship of CRS has been highlighted in this consensus classification on CRS [2]. In type-I [3] and type-III [4] CRS, multiple pathomechanisms, including the activation of the immune and neuroendocrine systems, may cause renal (in type-1 CRS) or cardiac (in type-III CRS) sequelae. However, information on the etiology and the course of CRS (whether or not the CRS occurred in an acute or chronic fashion) may be lacking. When serial laboratory tests on serum creatinine, cystatin C, brain natriuretic peptide, and N-terminal pro-brain natriuretic peptide and/or echocardiographic exam results are not available, assumptions need to be made regarding the acuity or chronicity of CRS. Specifically, without these laboratory and clinical exam results, the clinician cannot prove whether the disease process started in the heart (CRS type I and II) or in the kidneys (CRS type III and IV), or whether the disease process affected both organs concurrently (CRS type V). In order to support therapeutic decisions, the consensus definition needs to be revised. In addition, an updated CRS classification needs to be validated, as controversy about the validity and applicability of the current consensus classification of CRS still exists [5]. Here, a historical note on CRS is provided, and an updated classification of CRS is proposed.

## 2. Cardiorenal Syndrome: A Historical Note

In 1842, Carl F. W. Ludwig (Figure 1) proposed the hypothesis that urine is the result of a filtration process by the glomeruli promoted by the force of blood pressure [6]. Thus, the interplay of cardiac function (blood pressure) and renal function (urine production) was described for the first time. He further postulated that the final composition of the urine is the result of resorption processes taking place in the renal tubuli located downstream of the renal glomeruli. His work was published at a time when the physiological role of the kidneys was still being debated. At that time, a natural force (rather than blood pressure) was thought to be the driving force for urine production. The observation that blood pressure drives glomerular filtration was the starting point for a better understanding of the interplay between the heart and the kidneys. As a result of either a drop in blood pressure or bradycardia, the renal glomerular filtration capacity decreases. Likewise, as a result of less glomerular filtration, the ensuing hypervolemia may lead to acute heart failure or aggravate a chronic heart failure (CHF) condition. Thus, both the heart and kidneys are functionally dependent on each other. It is to Carl Ludwig’s credit that afferent autonomic innervation such as the aortic depressor nerve is seen in the context of circulatory control. In fact, Ludwig proved that the aortic depressor nerve enters the brain stem, affecting blood-pressure regulation. Thus, Carl Ludwig’s work laid the basis for further research on the autonomic control of circulation. Additionally, Ludwig greatly improved the methodology for physiology experiments by inventing an aortic blood-velocity meter. In summary, he greatly influenced the understanding of cardiorenal physiology as he stimulated physiology research both in Germany and internationally [7]. In 1931, Frank R. Winton demonstrated experimentally that an increase in renal-venous pressure by 24 mmHg is accompanied by a marked decrease in renal blood flow and a decrease in urine output. His research tested a hypothesis put forward by Carl Ludwig (1861) and by Rudolf P. H. Heidenhain (1881) that distal urine tubules are compressed by congested renal venules [8]. One hundred years after the groundbreaking work by Ludwig, Arthur C. Guyton described CRS as a combined cardiac and renal dysfunction aggravating the failure of both organs progressively [9]. Guyton contributed physiological research on the role of the peripheral circulation to blood-pressure regulation, which, in turn, affects renal function. During the last five decades, neurohumoral stimulation including sympathoactivation [10,11], oxidative stress [12], and microinflammation [13] have been proposed as pathomechanisms in CHF.

## 3. Clinical and Physiological Hallmarks of Cardiorenal Syndrome

Besides CHF, the activation of the sympathetic nervous system (SNS) [14] and chronically elevated inflammatory serum parameters [15] have been identified as clinical hallmarks of chronic kidney disease (CKD). Both sympathoactivation [3,4] and systemic inflammation [16] are considered to be key pathomechanisms in CRS as well. Hemodynamic abnormalities in CRS include venous congestion due to increased right ventricular filling pressure accompanied by a tricuspid annular dilatation with ensuing regurgitation [17]. In the latter pathomechanism, the centrally venous congestion impedes the renal venous blood flow and favors an intrarenal edema, which may further impede the intrarenal arterial perfusion. As proof, kidney sonography may demonstrate an attenuated arterial intrarenal perfusion being absent in the periphery of the kidneys. Clearly, in CRS, an arterial underfilling mechanism due to renal hypoperfusion, e.g., in heart failure with reduced ejection fraction and/or during hypotension, may represent an alternative explanation as to why renal function may cease in CRS. In fact, venous congestion due to increased right ventricular filling pressures, e.g., in diastolic left ventricular hypertrophy with left ventricular diastolic dysfunction, and renal hypoperfusion during hypotension or during periods of cardiac decompensation may lead to acute kidney injury with or without a pre-existing chronic kidney disease in CRS. As a proof of concept, once renal venous congestion alleviates—e.g., following the paracentesis of ascites—renal function improves readily [18]. Likewise, when renal venous congestion is diagnosed in clinical medicine, the use of loop diuretics regularly improves both venous congestion and renal function. Point-of-care ultrasound of abdominal veins including portal-vein flow pattern and inferior vena cava size [19], the sonographic detection of extravascular lung fluid [20], and the determination of Doppler-sonographic intrarenal venous flow patterns [21] may help direct therapy to achieve less venous congestion and an augmented renal function in hypervolemic CRS. Euvolemia, or a steady state with the least possible venous congestion, is to be maintained by achieving a low maintenance dose of loop diuretics by adjusting the recommended daily intake of water and by lowering salt intake to 2–3 g per day. Once renal dysfunction due to either venous congestion or renal arterial hypoperfusion occurs, cardiac function may deteriorate. Conversely, once renal function resumes, cardiac tricuspid annular dilatation may improve due to less right ventricular filling pressure. As a caveat, accidental hypovolemia due to polyuria during renal recovery, hyperglycemia-related polyuria in diabetes mellitus, or a prolonged use of high-dose loop-diuretics may lead to hypovolemic shock with attenuated renal perfusion and/or to acute cardiac failure. Figure 2 summarizes hyper- and hypovolemia-related pathomechanisms in CRS. In CRS, due to valvular heart disease, because of both signs and symptoms, hemodynamics and therapeutic implications may differ. Therefore, a differentiation between valvular and nonvalvular CRS is needed, if therapy stratification is the goal. Besides changes in hemodynamics, both anemia [22] and systemic inflammation [16] represent unifying final pathomechanisms in CRS.

## 4. Signs and Symptoms of Underlying Chronic Kidney Disease and Chronic Heart Failure

As outlined in Table 1, both CHF and advanced CKD share several signs, symptoms, and laboratory key findings. Based on the clinical exam, it is a challenge to determine the underlying etiology of CRS—e.g., whether acute or chronic heart failure preceded an acute or chronic kidney injury. Peripheral edema, pulmonary venous congestion, and interstitial pulmonary edema are commonly detected in advanced CKD, in CHF, or in the combination thereof, CRS. As for the pathomechanism, an activated SNS is regularly found both in advanced CKD [14] and in CHF [23]. SNS activation is governed by the afferent loop of cardiovascular reflexes and by the brain renin–angiotensin system [24]. The renin–angiotensin system in the brain, the renin–angiotensin–aldosterone system (RAAS) in the periphery, and the SNS are all interrelated: plasma angiotensin 2 may contribute to hypothalamic SNS activation at anatomic loci where a tight brain–blood barrier is lacking [25]. Likewise, an activated SNS may lead to renin release from juxtaglomerular cells via the sympathetic renal nerves [26]. In CKD, renin activation leading to RAAS activation is a hallmark [27]. In addition, hyperkalemia independently leads to aldosterone activation in advanced CKD. Conversely, the treatment of hyperkalemia using the potassium binder Patiromer has the potential to lower plasma aldosterone in advanced CKD [28]. Underlying sarcopenia or protein-energy wasting has been recognized for both CHF and ESRD. Hypoalbuminemia, which is regularly found in both CHF [29] and end-stage renal disease (ESRD) [30] is associated with an increased mortality. As for ESRD, nutritional efforts were unable to prevent the prevalent catabolism [31]. The paradoxical association between low levels of low-density lipoprotein cholesterol and mortality has been described both in CHF [32] and ESRD [33]. Even though a causality was not proven, the pathomechanism may relate to an impaired liver function comprising an impaired biosynthesis of transport proteins both in CHF and in CRS, as 47% of ESRD patients included in this prospective observational study had CHF comorbidity [33]. Lastly, anemia has associations with both CHF [34] and CKD [35]. While a decreased erythropoietin production occurs in advanced CKD [36], both early stages of CHF [37] and advanced CKD [38] are associated with iron deficiency due to increased hepcidin plasma levels down-regulating enteral iron absorption. Systemic inflammation represents a candidate pathomechanism for hepcidin activation in both CHF [39] and advanced CKD [15].

## 5. Updated Classification of Cardiorenal Syndrome

In retrospect, the older classification differentiating between severe and non-severe CRS [9] conveyed prognostic information when CRS is coined “severe”. For clinicians, this differentiation implies that immediate action is needed to avoid a vicious cycle towards progressive cardiac and renal failure and death. In contrast, the current consensus classification of CRS [1] highlights the complexity of the underlying causes; however, it lacks therapeutic or prognostic implications. Here, the proposed update of the consensus classification of CRS specifies whether CRS is acute or chronic, whether a valvular or nonvalvular heart disease is present and whether CRS associates with hyper- or hypovolemia (Figure 3). The first point, the descriptive information on an acute or chronic course of disease, rather simplifies the current consensus classification. CRS types I, III, and V may be regarded as acute CRS and types II, IV, and V as chronic CRS. Etiologic information on valvular and nonvalvular heart disease is necessary to direct therapeutic decisions. Valvular heart disease may be subjected to a correctional cardiologic procedure or to cardiac surgery. Conversely, nonvalvular heart disease as a component of CRS may be subjected to evidence-based medical therapies. Lastly, if euvolemia is absent, information on a prevalent hyper- or hypovolemia needs to be provided to further direct medical therapy.

## 6. Valvular versus Nonvalvular Cardiorenal Syndrome

To date, outcome data on valvular versus nonvalvular CRS are lacking. A comprehensive body of evidence exists for valvular heart disease with and without reduced left ventricular ejection fraction. Specifically, both surgical and transaortic-aortic valve implantation (TAVI) [40,41] are superior to medical therapy in symptomatic aortic stenosis. Likewise, in patients with aortic stenosis and reduced ejection fraction, TAVI is superior to medical therapy [42]. In the COAPT study, a randomized clinical trial of patients with functional mitral regurgitation and heart failure [43], the transcatheter mitral-valve repair was shown to be superior to medical therapy. The surgical treatment of asymptomatic severe mitral regurgitation was shown to be superior to medical therapy as well [44]. In symptomatic severe tricuspid regurgitation, a large non-randomized study showed the benefit of interventional valve repair therapy [45]. In nonvalvular CHF, the European Society of Cardiology recommends the use of five drug classes for heart failure with a reduced ejection fraction [46]. Therapy of heart failure with preserved ejection fraction has been updated as well: the sodium-glucose-transporter-2 inhibitor Empagliflozin has become a first-line therapy in patients with and without type-2-diabetes comorbidity [47]. Dedicated randomized clinical trials on valvular or nonvalvular CRS are needed to provide evidence for either subgroup of CRS patients.

## 7. Hypervolemic versus Hypovolemic Cardiorenal Syndrome

Intravascular and interstitial fluid volume expansion due to cardiac congestion and/or due to renal failure defines the clinical term “hypervolemia”, which characterizes many patients with CHF [48] and CRS [49]. An existing volume overload congestion may turn into a symptomatic clinical congestion requiring hospitalization. In hemodialysis patients, hypervolemia is associated with heparin-binding growth factor midkine release [50]. The underlying reason for cytokine activation in both CHF [39] and in CRS [16] is likely gut-edema predisposing to enteral toxin translocation. As clinical signs, the presence of peripheral edema and ana sarka supports the diagnosis of a hypervolemic CRS, while the absence of peripheral edema and signs of exsiccosis and/or arterial hypotension are in line with a hypovolemic CRS. Besides arterial hypotension, bradycardia has been reported as a cause for heart failure in the elderly [51] and, in a first case report, as a possible cause for acute cardiorenal syndrome [52]. As for CRS, the term “bradycardia, renal failure, atrioventricular blockade, shock, and hyperkalemia (BRASH) syndrome” has been coined to describe this pathomechanism [53]. Therefore, in the updated classification of CRS, the BRASH syndrome is considered to be a hypovolemic CRS.

## 8. Summary

The updated classification of CRS maintains the etiologic information introduced in the existing consensus classification (type I-V). However, emphasis is laid upon the course of disease, i.e., acute or chronic CRS. Verified etiologic information may be added by the correct type of CRS, and unverified etiologic information should be omitted. Adding clinical hallmarks (valvular or nonvalvular CRS / hyper- or hypovolemic CRS) allows for improved clinical decision making and opens up more research opportunities. Further validation studies are needed.

## Figures and Tables

**Figure 1 jcm-11-02896-f001:**
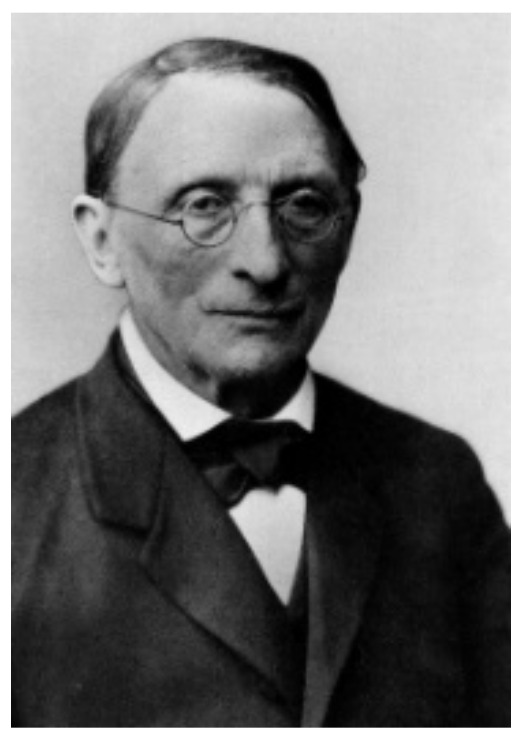
Title: Carl Ludwig (1816–1895), a pioneer of physiology, presented a new concept of renal function and cardiorenal interaction. Legend: Picture downloaded from https://research.uni-leipzig.de/agintern/UNIGESCH/ug175.htm (accessed on 17 May 2022).

**Figure 2 jcm-11-02896-f002:**
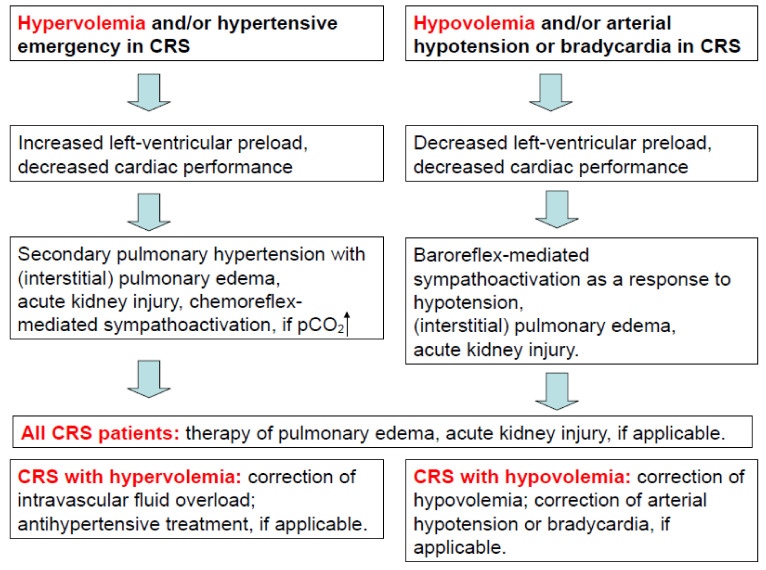
Title: Hypervolemia- and hypovolemia-related pathomechanisms in cardiorenal syndrome. Legend: Cardiovascular and autonomic-nervous-system-related and pathomechanisms in hypervolemic or hypovolemic CRS. Adapted with permission from reference [16]. (Copyright 2018: Linhart et al. ESC Heart Failure published by John Wiley & Sons Ltd on behalf of the European Society of Cardiology, Creative Commons Attribution-NonCommercial License).

**Figure 3 jcm-11-02896-f003:**
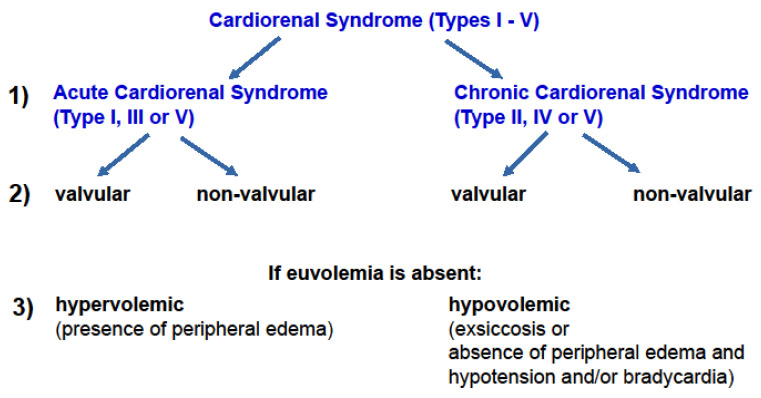
Title: A proposed new classification of cardiorenal syndrome. Legend: An upgraded classification of CRS requires clinical information that further specifies current consensus classification of CRS [1]. Aside from etiology and time course of CRS, clinical hallmarks such as the volemic state are considered.

**Table 1 jcm-11-02896-t001:** Clinical characteristics of chronic kidney disease and chronic heart failure.

	Chronic Kidney Disease (KDIGO G4-G5_nonD_)	Chronic Heart Failure (NYHA III-IV)
Peripheral edema	+	+
Pulmonary venous congestion	(+)	+
Interstital pulmonary edema	(+)	(+)
Sympathoactivation	+	+
Renin–angiotensin–aldosterone activation	+	+
Hypoalbuminemia	(+)	(+)
Cholesterol paradox	+	+
Anemia	+	+
Microinflammation	+	+

## Data Availability

Not applicable.

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
