# Peer review of "Cardiorenal Syndrome: An Updated Classification Based on Clinical Hallmarks"

_jcm, 2022, doi:10.3390/jcm11102896_

Round 1

Reviewer 1 Report

the author in this mini-review point to make an update of the Cardiorenal syndrome classification introducing informations on the presence or absence of valvular heart disease, acute or chronic disease and hyper or hypovolemia-associated to CRS.

Despite the paper is really interesting and well written, there are some little issues that should be addressed :

1- in tha paragraph 3: clinical and physiological hall marks of cardiorenal syndrome. Is there any chance to introduce a little table or image to summarize the points?

2- at page 4 line 128 provide the defition of abbraviation of SNS. Perhaps there was a typing error, because in the abbreviations section there is.

3- It could be interesting introduce these papers in the references:

Advances in the pathogenesis of cardiorenal syndrome type 3.

Clementi A, Virzì GM, Brocca A, de Cal M, Pastori S, Clementi M, Granata A, Vescovo G, Ronco C. Oxid Med Cell Longev. 2015;2015:148082. doi: 10.1155/2015/148082

The hemodynamic and nonhemodynamic crosstalk in cardiorenal syndrome type 1.

Virzì GM, Clementi A, Brocca A, de Cal M, Vescovo G, Granata A, Ronco C. Cardiorenal Med. 2014 Aug;4(2):103-12. doi: 10.1159/000362650.

Heart-kidney crosstalk and role of humoral signaling in critical illness.

Virzì G, Day S, de Cal M, Vescovo G, Ronco C. Crit Care. 2014 Jan 6;18(1):201. doi: 10.1186/cc13177  

Author Response

Reviewer #1:

the author in this mini-review point to make an update of the Cardiorenal syndrome classification introducing informations on the presence or absence of valvular heart disease, acute or chronic disease and hyper or hypovolemia-associated to CRS. Despite the paper is really interesting and well written, there are some little issues that should be addressed :

1- in the paragraph 3: clinical and physiological hall marks of cardiorenal syndrome. Is there any chance to introduce a little table or image to summarize the points?

Thank you for this important comment. Now, Figure 2 summarizes hypovolemia- and hypervolemia-related hallmarks of cardiorenal syndrome. In addition, the text was ammended accordingly:

Old (Page 6, lines 1-4): „As a caveat, accidental hypovolemia due to polyuria during renal recovery, due to hyperglycemia in diabetes mellitus or due to a prolonged use of high-dose loop-diuretics may lead to hypovolemic shock with attenuated renal perfusion and/or to cardiac failure due to arterial hypotension and less left-ventricular preload.“

New (Page 6, lines 10-14): „As a caveat, accidental hypovolemia due to polyuria during renal recovery, due to hyperglycemia-related polyuria in diabetes mellitus or due to a prolonged use of high-dose loop-diuretics may lead to hypovolemic shock with attenuated renal perfusion and/or acute cardiac failure. Figure 2 summarizes hypervolemia- and hypovolemia - related pathomechanisms in CRS.

2- at page 4 line 128 provide the defition of abbraviation of SNS. Perhaps there was a typing error, because in the abbreviations section there is.

The abbreviation SNS was defined when it was used for the first time on Page 5, first line.

3- It could be interesting introduce these papers in the references:

Clementi A, Virzì GM, Brocca A, de Cal M, Pastori S, Clementi M, Granata A, Vescovo G, Ronco C. Advances in the pathogenesis of cardiorenal syndrome type 3. Oxid Med Cell Longev. 2015;2015:148082. doi: 10.1155/2015/148082

Virzì GM, Clementi A, Brocca A, de Cal M, Vescovo G, Granata A, Ronco C. The hemodynamic and nonhemodynamic crosstalk in cardiorenal syndrome type 1. Cardiorenal Med. 2014 Aug;4(2):103-12. doi: 10.1159/000362650.

Virzì G, Day S, de Cal M, Vescovo G, Ronco C. Heart-kidney crosstalk and role of humoral signaling in critical illness. Crit Care. 2014 Jan 6;18(1):201. doi: 10.1186/cc13177

Thank you for this point. These references were included in the manuscript.

Reviewer 2 Report

In this review, the author reviews the current classification system for cardiorenal syndrome (CRS), it’s pathophysiology and some of the weaknesses with the current system. The author then proposes a need to specify when valvular disease contributes to CRS and whether the patient’s state is hypervolemia versus hypovolemia.

The biggest concern for this review and proposed classification system is I do not feel the author makes a strong enough argument for their proposed classification system of both valvular/non-valvular and hyper/hypovolemia. There are a few sentences on tricuspid valve changes with fluid overload, but what about aortic stenosis or mitral regurgitation? Why does the pathophysiology of these states differ than non-valvular? Does the author have any evidence to back up importance of a valvular specific like from COAPT or other studies? Similarly hyper/hypovolemia only has a few sentences in pathophysiology without a clear delineation why this needs to be differentiated. Frankly, I do not think anyone consider hypovolemia CRS, but just hypovolemic AKI, thus not sure this differentiation is needed. Overall, the review would benefit from two new sections between current section 4 and 5 with one section addressing importance of distinguishing valvular versus non-valvular with evidence to back up this split and another section describing importance of hypervolemia versus hypovolemia (though again, I do not think there is really a need for this categorization).

My other concerns as follows.

Major:

-While CRS is most often in the context of heart failure, by no means is it heart failure specific but more a cardiovascular condition leading to kidney disease and vice versa. As is the authors primary goal to describe valvular CRS as a different type of CRS, this fits that it is not HF specific. Additionally, it should be explicitly stated type 5 is another disease state promoting injury to both organs, such as sepsis or diabetes. Thus, on line 167 where the author lumps CRS type 5 with ‘chronic’ state is not accurate as type 5 could be an acute or chronic state. I would suggest type 5 stay it’s own category ‘non-cardiac, non-renal CRS’ of sorts.

-The author explains weakness of the current classification system throughout text, but this should be discussed a bit more in-depth in the introduction. Such as the author notes that it is difficult to determine acuity and chronicity of disease, which organ impacts the other, and current classification does not direct treatment.

-would also mention early work of Winton in 1931 PMID: 16994199

-for pathophysiology, the statement “In the latter pathomechanism, the centrally venous congestion impedes the renal venous blood flow, favors an intrarenal edema which may further impede the intrarenal arterial perfusion.” Focuses on arterial flow, but venous flow patterns also shown to be altered and predictive (PMID: 33477157, PMID: 33118672, PMID: 31630601)

Minor:

Abstract has typo with repetition of chronic kidney disease: “clinical hallmarks of both chronic kidney disease and chronic kidney disease are discussed”

Line 64 with typo or missing word, confusing sentence: “In fact, the he proved that the aortic”

Author Response

Reviewer #2:

In this review, the author reviews the current classification system for cardiorenal syndrome (CRS), it’s pathophysiology and some of the weaknesses with the current system. The author then proposes a need to specify when valvular disease contributes to CRS and whether the patient’s state is hypervolemia versus hypovolemia.

The biggest concern for this review and proposed classification system is I do not feel the author makes a strong enough argument for their proposed classification system of both valvular/non-valvular and hyper/hypovolemia. There are a few sentences on tricuspid valve changes with fluid overload, but what about aortic stenosis or mitral regurgitation? Why does the pathophysiology of these states differ than non-valvular? Does the author have any evidence to back up importance of a valvular specific like from COAPT or other studies? Similarly hyper/hypovolemia only has a few sentences in pathophysiology without a clear delineation why this needs to be differentiated. Frankly, I do not think anyone consider hypovolemia CRS, but just hypovolemic AKI, thus not sure this differentiation is needed. Overall, the review would benefit from two new sections between current section 4 and 5 with one section addressing importance of distinguishing valvular versus non-valvular with evidence to back up this split and another section describing importance of hypervolemia versus hypovolemia (though again, I do not think there is really a need for this categorization).

Thank you for these important points that may help to describe the proposed update of CRS classification more clearly. There is consensus that medical therapy is not the first choice in symptomatic valvular heart disease. Likwewise, patients presenting with hypovolemic or hypervolemic CRS need to be treated differently. As suggested, the two new sections were introduced.

Old (Page 8, last 9 lines): „In hemodialysis patients, hypervolemia associates with heparin-binding growth factor midkine release [35]. Aside from presence or absence of peripheral edema and ana sarka, arterial hypotension and/or episodes of bradycardia are included in the phenotype of a hypovolemic CRS. Since the first report of bradycardia-associated acute cardiorenal syndrome [36], the term “bradycardia, renal failure, atrioventricular blockade, shock, and hyperkalemia (BRASH)” syndrome has been coined to describe this pathomechanism [37]. In the updated classification of CRS, the BRASH syndrome is represented within the framework of a hypovolemic CRS.“

New (Page 8, 2nd paragraph – Page 9, last line):

Valvular versus nonvalvular cardiorenal syndrome

To date, outcome data on valvular versus nonvalvular CRS are lacking. A comprehensive body of evidence exists for valvular heart disease with and without reduced left ventricular ejection fraction. Specifically, both surgical and transaortic aortic valve implantation (TAVI) [42-43] is superior to medical therapy in symptomatic aortic stenosis. Likewise, in patients with aortic stenosis and reduced ejection fraction, TAVI is superior to medical therapy [44]. In the COAPT study, a randomized clinical trial of patients with functional mitral regurgitation and heart failure [45], the transcatheter mitral-valve repair was shown to be superior to medical therapy. Surgical treatment of asymptomatic severe mitral regurgitation was shown to be superior to medical therapy as well [46]. In symptomatic, severe tricuspid regurgitation, a large non-randomized study showed a benefit for interventional valve repair therapy [47]. In nonvalvular CHF, the European Society of Cardiology recommends the use of 5 drug classes in heart failure with reduced ejection fraction [48]. Therapy of heart failure with preserved ejection fraction has been updated as well: the sodium-glucose-transporter-2 inhibitor empagliflozin has become a first-line therapy in patients with and without type-2-diabetes comorbidity [49]. Dedicated randomized clinical trials on valvular or nonvalvular CRS are needed to provide evidence for either subgroup of CRS patients.

Hypervolemic versus hypovolemic cardiorenal syndrome

Intravascular and interstitial fluid volume expansion due to cardiac congestion and/or due to renal failure defines the clinical term “hypervolemia” characterizing many patients with CHF (50) and CRS (51). An existing volume overload congestion may turn into a symptomatic clinical congestion requiring hospitalization. In hemodialysis patients, hypervolemia associates with heparin-binding growth factor midkine release [52]. The underlying reason for cytokine activation both in CHF [53] and in CRS [16] likely is gut-edema predisposing to enteral toxin translocation. As clinical signs, the presence of peripheral edema and ana sarka supports the diagnosis of a hypervolemic CRS, while absence of peripheral edema and signs of exsiccosis and/or arterial hypotension are in line with a hypovolemic CRS. Besides arterial hypotension, bradycardia has been reported as a cause for heart failure in the elderly [54] and, in a first case report, as a possible cause for acute cardiorenal syndrome [55]. As for CRS, the term “bradycardia, renal failure, atrioventricular blockade, shock, and hyperkalemia (BRASH) syndrome” has been coined to describe this pathomechanism [56]. Therefore, in the updated classification of CRS, the BRASH syndrome is considered to be a hypovolemic CRS.“

My other concerns as follows.

Major:

-While CRS is most often in the context of heart failure, by no means is it heart failure specific but more a cardiovascular condition leading to kidney disease and vice versa. As is the authors primary goal to describe valvular CRS as a different type of CRS, this fits that it is not HF specific. Additionally, it should be explicitly stated type 5 is another disease state promoting injury to both organs, such as sepsis or diabetes. Thus, on line 167 where the author lumps CRS type 5 with ‘chronic’ state is not accurate as type 5 could be an acute or chronic state. I would suggest type 5 stay it’s own category ‘non-cardiac, non-renal CRS’ of sorts.

Indeed, CRS is not heart-failure specific and we maintained this view from the current consensus classification (reference 1) throughout the manuscript. In addition, we corrected the presentation of CRS type V as being an acute or a chronic CRS. We gratefully follow your suggestion in highlighting CRS type V as its own category. The manuscript and the figure 3 were changed accordingly.

Old (Page 3, lines 5-9):

Five CRS types were described: type-I CRS presenting as an acute cardiac failure leading to an acute renal failure, type-II CRS presenting as chronic cardiac failure leading to a chronic renal failure, type-III CRS presenting as acute kidney injury aggravating heart failure, type-IV CRS presenting as chronic kidney failure aggravating heart failure, and type-V CRS presenting as a concurrent acute cardiac and renal failure or injury [1].“

New (Page 3, lines 5-11):

Five CRS types were described: type-I CRS presenting as an acute cardiac failure leading to an acute renal failure, type-II CRS presenting as chronic cardiac failure leading to a chronic renal failure, type-III CRS presenting as acute kidney injury aggravating heart failure, type-IV CRS presenting as chronic kidney failure aggravating heart failure, and type-V CRS presenting as a concurrent acute or chronic cardiac and renal failure or injury [1]. CRS type V represents an own category, as non-cardiac and non-renal conditions such as diabetes mellitus or sepsis are the offending causes.“

-The author explains weakness of the current classification system throughout text, but this should be discussed a bit more in-depth in the introduction. Such as the author notes that it is difficult to determine acuity and chronicity of disease, which organ impacts the other, and current classification does not direct treatment.

As for the difficulty to determine acuity and chronicity of disease, which organ impacts the other, the introduction was modified. As for the controversy about the validity and applicability of the current classification, the cited reference 5 (last sentence of introduction) was mentioned.

Old (Page 3, lines 12-17): „When serial laboratory and/or echocardiographic exams results are not available, assumptions need to be made towards etiology of CRS and towards the course of disease. Specifically, the clinician cannot prove whether the disease process started in the heart (CRS type I and II) or in the kidneys (CRS type III and IV), or whether the disease process affected both organs concurrently (CRS type V).“

New (Page 3, lines 16-22): „When serial laboratory tests on serum creatinine, cystatin C, brain natriuretic peptide, N-terminal pro-brain natriuretic peptide and/or echocardiographic exam results are not available, assumptions need to be made towards the acuity or chronicity of CRS. Specifically, without these laboratory and clinical exam results, the clinician cannot prove whether the disease process started in the heart (CRS type I and II) or in the kidneys (CRS type III and IV), or whether the disease process affected both organs concurrently (CRS type V).“

- would also mention early work of Winton in 1931 PMID: 16994199

Thank you for this reference that perfectly fits into the paragraph „Cardiorenal syndrome: a historical note“.

New (Page 4, lines 21-25): „ In 1931, Frank R. Winton demonstrated experimentally that an increase of renal-venous pressure by 24 mmHg is accompanied by a marked decrease of renal blood flow and a decrease of urine output. His research tested a hypothesis put forward by Carl Ludwig (1861) and by Rudolf P. H. Heidenhain (1881) that distal urine tubules were compressed by congested renal venules [8].“

-for pathophysiology, the statement “In the latter pathomechanism, the centrally venous congestion impedes the renal venous blood flow, favors an intrarenal edema which may further impede the intrarenal arterial perfusion.” Focuses on arterial flow, but venous flow patterns also shown to be altered and predictive (PMID: 33477157, PMID: 33118672, PMID: 31630601)

The suggested references are adding a strong point, thanks.

New (Page 5, last line – Page 6, lines 1-4): „Point-of-care ultrasound of abdominal veins including portal-vein flow pattern and inferior vena cava size [19], sonographic detection of extravascular lung fluid [20] and determination of Doppler-sonographic intrarenal venous flow patterns [21] may help direkt therapy to achieve less venous congestion and an augmented renal function in hypervolemic CRS.“

Minor:

Abstract has typo with repetition of chronic kidney disease: “clinical hallmarks of both chronic kidney disease and chronic kidney disease are discussed”

Thanks. The typo was rectified:

Old (Page 2, Abstract, 1st sentence): „Cardiorenal syndrome (CRS) is defined as a progressive, combined cardiac and renal dysfunction. In this mini-review, a historical note on CRS is presented, pathomechanisms and clinical hallmarks of both chronic kidney disease and chronic kidney disease are discussed and an updated classification of CRS is proposed.“

New (Page 2, Abstract, 1st sentence): „Cardiorenal syndrome (CRS) is defined as a progressive, combined cardiac and renal dysfunction. In this mini-review, a historical note on CRS is presented, pathomechanisms and clinical hallmarks of both chronic heart failure and chronic kidney disease are discussed and an updated classification of CRS is proposed.“

Line 64 with typo or missing word, confusing sentence: “In fact, the he proved that the aortic”

The typo was corrected.

Old (Page 4, first paragraph, line 12): „In fact, the he proved that the aortic depressor nerve enters the brain stem affecting blood-pressure regulation.”

New (Page 4, first paragraph, line 15): „In fact, Ludwig proved that the aortic depressor nerve enters the brain stem affecting blood-pressure regulation.“

Reviewer 3 Report

Dear Author,

the review is well written, even if it's focused on a topic with reduced interest in recnt years

My suggestion concerns figure 2 as it's the "beating heart" of your paper; please make it more clear and readable using, for example the smart art graphic solitions. 

Author Response

Reviewer #3:

Dear Author, the review is well written, even if it's focused on a topic with reduced interest in recnt years

My suggestion concerns figure 2 as it's the "beating heart" of your paper; please make it more clear and readable using, for example the smart art graphic solutions.

Thank you for this point. The former figure 2 (now figure 3) has been updated accordingly.

Round 2

Reviewer 3 Report

thanks a lot for your revision

no other comments